# Integrated multi-omics analysis reveals necroptosis-related biomarker BIRC3 for early diagnosis and therapeutic targeting in preeclampsia

Qingxia Lin[1�*/], Peifeng Huang[2�)], Youhong Kang[1], Yanfeng Lu[1]*, Guili Shi[1]*

**1** Department of Obstetrics, Quanzhou Women and Children's Hospital, Quanzhou, China, **2** The Second Affiliated Hospital of Fujian Medical University, Quanzhou, China

☯ These authors contributed equally to this work.
* secret183@163.com (GLS); 961324494@qq.com (YFL)

## Abstract

### Background

Preeclampsia (PE) is a life-threatening pregnancy disorder lacking reliable early biomarkers. While apoptosis is implicated in PE pathogenesis, the role of regulated necrotic cell death (necroptosis) remains poorly understood. This study aimed to identify necroptosis-related biomarkers, and further provide the potential natural compounds for PE with virtual screening.

### Methods

Public datasets (GSE66273 for training set; GSE44711 for validation set; GSE173193 for single-cell RNA-seq) were analyzed. Differentially expressed genes (DEGs) were screened using limma ($|\log2FC| > 1$, $P < 0.05$). Necroptosis-related genes overlapped with DEGs to identify key candidates. Subsequent analyses included machine learning, protein-protein interaction (PPI) network construction, and immune infiltration profiling. Single-cell RNA sequencing data (GSE173193) was utilized to localize BIRC3 expression at the cellular level. Transcription factors, microRNAs, and RNA-binding proteins associated with BIRC3 are also identified. Finally, Molecular docking predicted therapeutic drugs targeting hub genes.

### Results

The analysis of the GSE66273 dataset identified 367 DEGs. Intersection with necroptosis-related genes revealed 3 necroptosis-related DEGs (NRDEGs), from which BIRC3 was prioritized as hub gene through PPI networks and machine learning (random forest). BIRC3 demonstrated significant diagnostic potential in the discovery cohort (AUC = 0.933) and maintained strong performance in the independent

**Data availability statement:** All the data were collected and downloaded from GEO database (GSE66273, GSE173193, GSE44711).

**Funding:** The author(s) received no specific funding for this work.

**Competing interests:** The authors have declared that no competing interests exist.

validation cohort (AUC = 0.844). Single-cell analysis revealed BIRC3 was predominantly expressed in immune lineages, particularly NK/T cells, with a significantly higher proportion of BIRC3-positive cells in PE placentas (p < 0.05). Immune microenvironment further analysis demonstrated significant dysregulation in PE. Finally, based on the BIRC3 structure, Withanolide D (−11.0 kcal/mol), Baicalin (−9.0 kcal/mol) and Mangostin (−8.3 kcal/mol) were screened.

## Conclusion

This comprehensive analysis implicates necroptosis in PE pathogenesis. BIRC3 is proposed as a novel diagnostic biomarker and therapeutic target, with multi-omics validation underscoring its role in immune dysregulation and placental dysfunction.

## 1. Introduction

Preeclampsia (PE) is a serious hypertensive disorder of pregnancy that typically manifests as new-onset hypertension and proteinuria after 20 weeks of gestation. It is a leading cause of maternal and perinatal morbidity and mortality worldwide, affecting approximately 5%−7% of all pregnancies [1,2]. The clinical progression of PE can be rapid and unpredictable, culminating in multi-organ dysfunction including eclampsia, HELLP syndrome, and placental abruption [3,4]. Despite its significant health burden, the definitive treatment remains the delivery of the placenta, often necessitating preterm birth with its associated complications. The absence of reliable early diagnostic biomarkers underscores an urgent need to elucidate the underlying pathogenic mechanisms, which is a prerequisite for developing predictive strategies and novel therapeutics.

PE is affected by a variety of factors. Currently, the etiology of PE is believed to be due to an insufficient nutrient supply to the spiral artery caused by trophoblast infiltration, which leads to insufficient spiral artery remodeling and eventually preeclampsia. In addition, disease progression is associated with changes in placental development, perfusion, and nutrient transport [5]. Changes at the molecular level are involved in inflammation, oxidative stress, apoptosis and endothelial cell dysfunction [6]. In recent years, increasing attention has been given to the role of the balance between the proliferation and apoptosis of cytotrophoblast cells in the normal development of the placenta. Once these functions are out of balance, syncytial trophoblast cells are driven to necrosis and rupture, and corresponding fragments are released, which may shed new light on the earliest mechanism of PE [7]. Many studies have reported a significant increase in the rate of cell apoptosis in the placenta of patients with PE [8,9].

Necroptosis is a regulated inflammatory cell death, distinct from apoptosis, ferroptosis, and pyroptosis [10–12]. It is initiated by death receptors leading to necrosome formation, a complex of RIPK1 and RIPK3. RIPK3 then phosphorylates MLKL, triggering its oligomerization, membrane translocation, and subsequent membrane rupture, which releases intracellular pro-inflammatory contents [13]. Recent studies

have shown that necroptosis is associated with inflammatory disease, neurodegenerative diseases, ischemic cardio-vascular and placental-related diseases [14–17]. Mounting evidence underscores its potential role in PE, where key molecular components of the necroptotic pathway demonstrate aberrant expression in placental tissues. For instance, the expression level of RIPK1 and p-MLKL are significantly elevated in PE placentas [18,19]. Furthermore, studies have revealed increased protein levels of the necroptosome components RIP1 and RIP3 in PE placental tissues, accompanied by MLKL activation and oligomerization, alongside observations of cytotrophoblast morphology exhibiting necrotic characteristics [20]. Importantly, the functional significance of this pathway is highlighted by the ability of necroptosis inhibitors (necrostatin-1 and GSK'872) to restore trophoblast viability [21]. Thus, necroptosis emerges as a critical nexus linking placental ischemia/hypoxia to the exaggerated systemic inflammatory state in PE. Nevertheless, a comprehensive landscape of the necroptosis-related gene expression profile, their functional significance, and their crosstalk with the dysregulated immune microenvironment in PE remains elusive, necessitating systematic investigation.

To address this knowledge gap, we leveraged an integrated multi-omics bioinformatics approach. This strategy is powerful for hypothesis generation from existing public genomic data, allowing for the systematic identification of high-confidence candidates and the construction of coherent biological narratives that can guide future targeted research. In the present study, we performed a systematic analysis of necroptosis-related differentially expressed genes (NRDEGs) between control placenta and preeclampsia placena based on the dataset GSE66273. We further explored the potential functional mechanisms and identified hub gene BIRC3 associated with necroptosis, while also examining the relationship between necroptosis and immune cell infiltration in PE. External validation of the BIRC3 was conducted using an independent sequencing dataset (GSE44711), and the diagnostic value was evaluated. Single-cell RNA sequencing (scRNA-seq) was employed to localize the expression of a key hub gene, BIRC3, within specific placental cell types. Additionally, molecular docking was performed to screen potential therapeutic compounds targeting the BIRC3. By applying this multi-faceted analytical approach, we aim to explore novel mechanistic insights into PE and to identify candidate biomarkers and therapeutic targets, thereby generating testable hypotheses for future research.

## 2. Materials and methods

### 2.1 Data acquisition

The GSE66273 dataset (5 preterm tacontrol placenta and 6 preterm preeclampsia placen datasets) and GSE44711 (8 Early Onset Preeclampsia placentas and 8 gestational age matched controls), were downloaded from the GEO database. The dataset was normalized through the normalizeBetweenArrays function of the "limma" package, and the data correction of each sample was visualized via a box plot. The workflow chart is displayed in Fig 1. The original studies associated with these datasets had received necessary ethical approval, as documented in their respective source publications.

### 2.2 Analysis of DEGs

Differential analysis for group comparisons was performed via the R/Bioconductor package "limma". The DEGs were screened using the criteria of a $|log2(FC)| > 1$ and a p value $< 0.05$ (detailed results in S2 Table). Principal component analysis (PCA), volcano plots, difference ranking plots and heatmap plots were generated via the 'ComplexHeatmap' and 'ggplot2' packages. Additionally, necroptosis-related genes were collected by searching for GeneCards Database [22] with "necroptosis" as the keyword, the genes with a relevance score greater than twice the median were selected (detailed results in S1 Table). Next, we defined NRDEGs by taking the intersection of DEGs, and the overlap was visualized with Venn diagrams.

### 2.3 Functional enrichment analysis

To understand the biological significance of the DEGs, we performed enrichment analysis. Gene Ontology (GO) and Kyoto Encyclopedia of Genes and Genomes (KEGG) analyses were performed via the R package clusterProfiler. GO terms

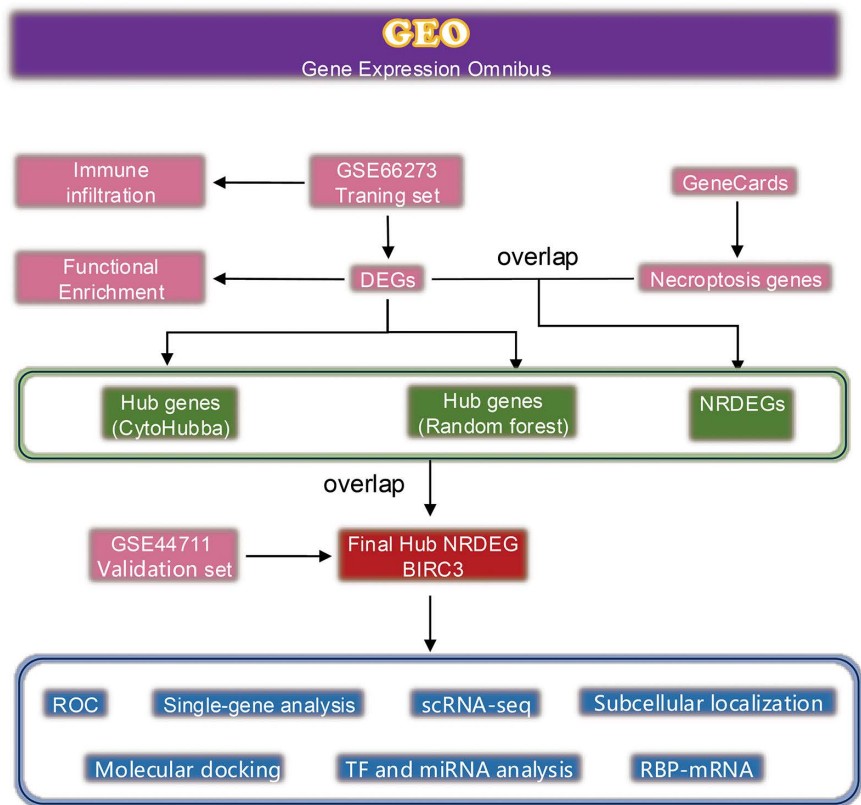

**Fig 1. Flow chart of the study design.**

and KEGG pathways with p values less than 0.05 were considered significant categories. Moreover, gene set enrichment analysis (GSEA) was performed via GSEA software (version 4.1.0).

## 2.4 Single-cell RNA sequencing (scRNA-seq) analysis

The raw single-cell RNA-seq data from the GSE173193 dataset were downloaded and processed for subsequent analysis. Quality control and data preprocessing were performed using the Seurat R package (v4.1.0). Low-quality cells were filtered out based on the following criteria: cells with fewer than 200 or more than 7500 detected genes, as well as cells with a mitochondrial content exceeding 20%. After quality filtering, 22976 high-quality cells were retained for downstream analysis. Read counts were normalized per cell using a scale factor of 10,000 and log-transformed. The data were then scaled using the ScaleData function in Seurat to regress out potential confounding sources of variation. The top 2,000 highly variable genes were selected for dimensionality reduction via principal component analysis (PCA). The top 30 principal components were used for cell clustering and visualization. Cell clusters were identified using the FindNeighbors and FindClusters functions (resolution = 0.05) in Seurat. Cell type annotation was performed by combining marker gene expression analysis via the FindAllMarkers function and automated cell type labeling using the SingleR package. Finally, the expression of BIRC3 across cell clusters and between sample groups was visualized using the FeaturePlot and DotPlot functions in Seurat. The difference in the proportion of BIRC3-positive cells between the two groups was compared using a Chi-square test, with a p-value of less than 0.05 considered statistically significant.

## 2.5 Immune infiltration analyses

To obtain a more robust assessment of immune infiltration, we adopted two complementary computational methods: xCell [23], single-sample gene set enrichment analysis (ssGSEA) and CIBERSORTx. The infiltration levels of immune and stromal cells were assessed using xCell, a deconvolution algorithm that calculates enrichment scores for a comprehensive panel of 64 cell types from gene expression data. This method utilizes a novel signature-based approach to accurately resolve a broad cellular spectrum, demonstrating improved performance in distinguishing closely related cell subsets. The pathway-centric view of gene set activation was assessed using ssGSEA. This computational algorithm calculates enrichment scores for predefined, cell-type-specific gene signatures in each placenta specimen, providing a quantitative measure of pathway activity. CIBERSORTx uses a support vector regression model and allows for sensitivity analysis with different signature matrices. This multi-method approach was employed to triangulate findings and enhance the robustness of the results, as different algorithms have unique strengths and potential biases.

## 2.6 Analysis of the PPI network and identification of hub genes

Genes participate in all aspects of life processes through interactions with each other. Clarifying the functional connection between genes in the disease state is highly important for understanding pathogenesis and screening key genes. PPI networks were constructed on the basis of NRDEGs by using the STRING database [24], with a minimum required interaction score of 0.4 (medium confidence). The resulting network was then visualized and analyzed in Cytoscape software [25].

## 2.7 Identification of diagnostic genes

To determine whether the hub NRDEG can be used as diagnostic biomarker of PE, the differences in gene expression between preeclampsia patients and preterm controls were visually displayed via violin plots. A ROC curve was subsequently used to determine the diagnostic value of hub NRDEG in PE.

## 2.8 Construction of TF-mRNA and miRNA–mRNA regulatory networks

The putative transcription factor (TF)–mRNA and miRNA–mRNA interaction networks for the hub gene BIRC3 were constructed using the NetworkAnalyst platform (https://www.networkanalyst.ca/). The platform employed the BETA Minus algorithm with thresholds to select high-confidence interactions: only peaks with a signal intensity < 500 and a regulatory potential score < 1 were included. miRNAs affect many biological processes through regulating the expression of target genes [26]. To investigate how miRNAs regulate hub NRDEG in PE, the miRNA–gene coregulatory network was obtained from the miRTarBase database, which is included in the NetworkAnalyst platform. miRNAs with the minimal network connectivity were identified. Then, the necroptosis-related TF-target gene regulatory interaction network was generated from the TRRUST dataset in the NetworkAnalyst platform. TFs of the BIRC3 were predicted within a minimal network.

## 2.9 Construction of an RBP–mRNA regulatory network

RNA-binding proteins (RBPs), which are involved in mRNA splicing, maturation, transport, and translation, are the dominant forces involved in posttranscriptional gene regulation [27]. The RBP-mRNA interaction network was predicted from the starBase of ENCORI (https://rnasysu.com/encori/index.php). Here, the RBP-RNA interactions supported by the binding sites of RBPs derived from CLIP-seq data were presented. All identified binding sites within 200 bp of a RBP will be defined as a RBP-binding cluster.

## 2.10 Potential therapeutic drug prediction

Hub NRDEG-related drugs were predicted through the drug signature database (DGIdb), which is included in the Enrichr platform [28].

## 2.11 Molecular docking

Molecular docking studies were performed to predict the binding mode and affinity of the compound(s) against the BIRC3 protein using the online web server CB-Dock2 (http://cao.labshare.cn/cb-dock2/), which is an automated, curve-based blind docking tool [1]. The molecular structures was retrieved from PubChem Compound (https://pubchem.ncbi.nlm.nih.gov/). The 3D coordinates of BIRC3 were downloaded from the PDB (http://www.rcsb.org/pdb/home/home.do). The prepared protein (.pdb) and ligand (.pdbqt) files were uploaded to the CB-Dock2 web interface with all water molecules excluded and polar hydrogen atoms were added.The docking calculation was conducted with the following parameters: ① Number of Cavities: 5; ② Docking Exhaustiveness: 8; ③ Maximum Output Modes: 9; ④ Box Size Adjustment: Automatic (default). The server automatically identified the top 5 potential binding pockets on the protein surface based on cavity detection algorithms. For each predicted cavity, the compound was flexibly docked, and 9 potential binding poses were generated, ranked by their predicted binding affinity (Vina Score, in kcal/mol).

## 3. Results

### 3.1 Identification of DEGs and functional enrichment analysis

We downloaded the dataset GSE66273 from the GEO database. Normalization was performed on the expression matrices of the dataset. Fig 2A shows the average normalized expression trend of the genes. The PCA indicated that the data of each sample were repeatable (Fig 2B). Next, we carried out differential gene expression analysis between preeclamptic placenta and preterm control placenta via the limma package. A total of 367 DEGs were ultimately screened, including 200 significantly upregulated genes and 167 significantly downregulated genes, which are visualized in the volcano plot and the heatmap (Fig 2C and 2D). The difference ranking plot revealed the top four upregulated (LEP, MIG7, ARHGEF4, and CGB) and downregulated genes (ASB2, CRYM, OPRK1, and SLC28A2) (Fig 2E).

We then conducted functional enrichment analysis to explore the functions associated with the upregulated genes. The GO analysis consists of three items: cellular component (CC), molecular function (MF), and biological process (BP). As clearly shown in Fig 3A, significant BPs were enriched mainly in positive regulation of leukocyte cell–cell adhesion (GO: 1903039), production of molecular mediators of the immune response (GO: 0002440), cell chemotaxis (GO: 0060326), and response to hypoxia (GO: 0001666). The GO enrichment analysis revealed enrichment of the MHC class II protein complex (GO: 0042613), COPII-coated ER to Golgi transport vesicle (GO: 0030134), and brush border membrane (GO: 0031526) was also detected in CCs (Fig 3B). Furthermore, enrichment of hormone activity (GO: 0005179), metalloendopeptidase inhibitor activity (GO: 0008191), and glycosaminoglycan binding (GO: 0005539) in MF (Fig 3C). Most importantly, KEGG pathway analysis revealed that immune- and inflammation-related pathways were enriched in PE patients (Fig 3D). To assess functional enrichment at the global level, we conducted GSEA. Consistent with the results of the KEGG and GO analyses, GSEA strongly revealed that the progression of PE is associated with hypoxia, inflammation, metabolic abnormalities and immune regulation (Fig 3E).

### 3.2 Identification of NRDEGs

We mapped the DEGs to the necroptosis gene set and identified 3 overlapping genes, of which 2 were upregulated and 1 were downregulated (Fig 4A, Table 1). The expression heatmaps of the 3 genes are presented in Fig 4B. Spearman correlations were used to further characterize the relationships among the 3 DEGs in the PE group (Fig 4C). To explore the functional processes related to the 3 NRDEGs, we carried out gene functional enrichment studies. GO analysis revealed that 3 NRDEGs are significantly enriched in pathways such as regulation of cysteine-type endopeptidase activity involved in apoptotic process, intracellular receptor signaling pathway, regulation of necroptotic process and endothelial cell chemotaxis (Fig 4D–4F). KEGG pathway enrichment showed that these DEGs are primarily involved in Apoptosis, Cortisol synthesis and secretion, NF-kappa B signaling pathway, TNF signaling pathway, and Necroptosis, which aligns with the

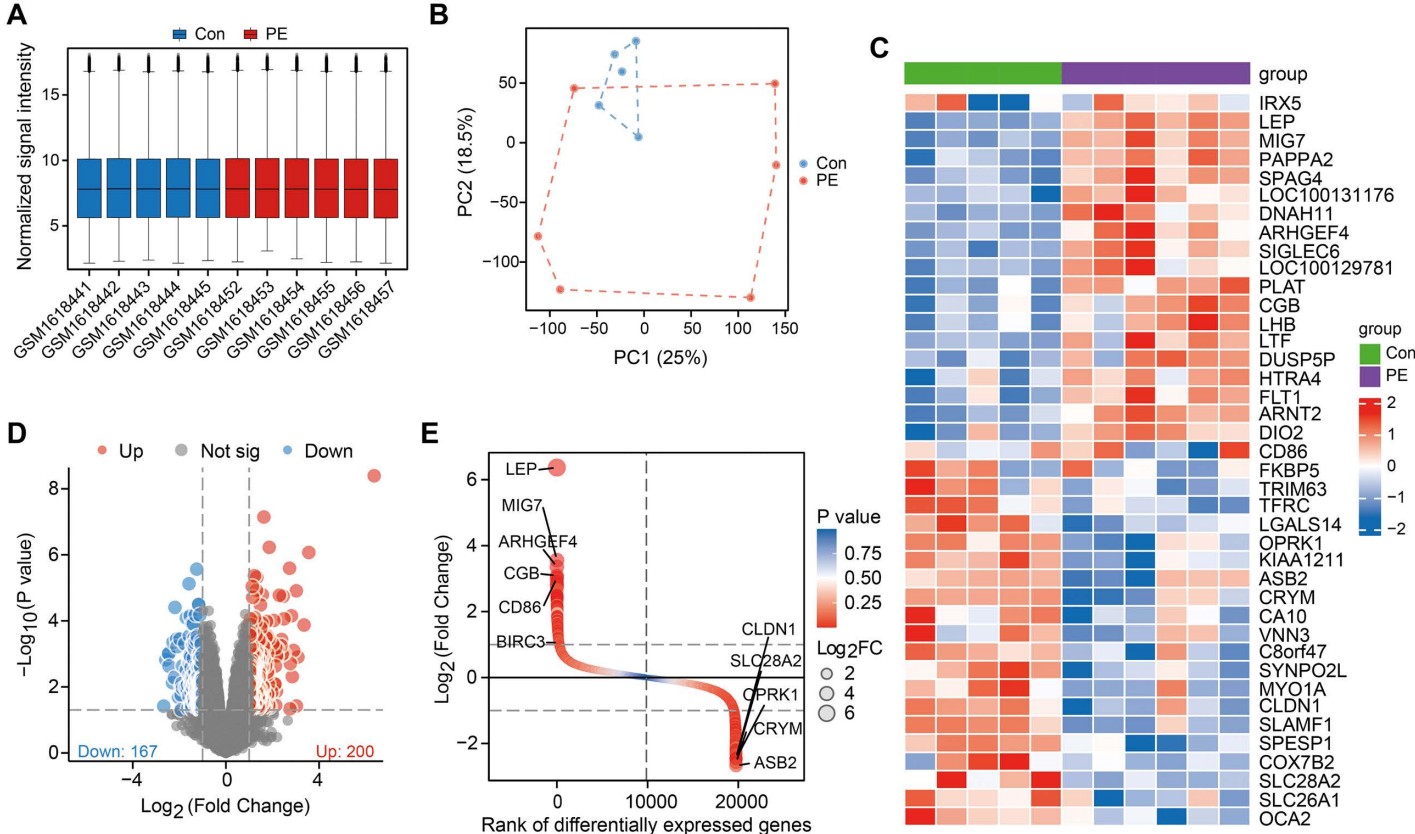

**Fig 2. Identification of differentially expressed genes (DEGs) between preterm preeclampsia placenta and preterm control placenta. (A)** The boxplot shows the data corrected for each sample (preterm preeclampsia placenta: n = 6; preterm control placenta: n = 5). There was no significant difference in the median or the upper and lower quartiles. **(B)** Principal component analysis (PCA) showing segregation between the two groups. **(C)** Heatmap showing the expression patterns of the DEGs between the two groups. **(D)** Volcano map of all the DEGs between preeclamptic placenta and preterm control placenta. Red dots: upregulated genes; blue dots: downregulated genes. **(E)** The difference ranking plots visualize the DEGs between the two groups.

established inflammatory pathogenesis of PE. Given the limited number of NRDEGs, the functional enrichment analysis should be interpreted as preliminary and exploratory. Nonetheless, it offered initial insights into the potential biological themes associated with these genes.

### 3.3 Identification of hub NRDEG and diagnostic feature biomarker

To definitively identify a core regulator, we employed a multi-step strategy to pinpoint hub genes. First, PPI network was constructed for upregulated DEGs via the STRING database. Then, Cytoscape software was used to select the hub genes on the basis of the degree, EPC, stress, and eccentricity algorithms (Fig 5A–5D). The intersection of these results yielded nine preliminary candidate hub genes (LTF, KIT, AREG, NOD2, CXCR3, BIRC3, MMP9, LEP, and BCL6) (Fig 5E). Concurrently, a Random Forest model was used to rank genes by their importance for classifying PE and control samples. The top 20 feature genes from the random forest analysis are displayed in Fig 5F and 5G. The intersection of these two independent candidate lists with the necroptosis-related DEGs (NRDEGs) uniquely identified BIRC3 (Fig 5H). The convergence of BIRC3 as a top candidate through both network-based and machine-learning approaches underscored its potential critical role.

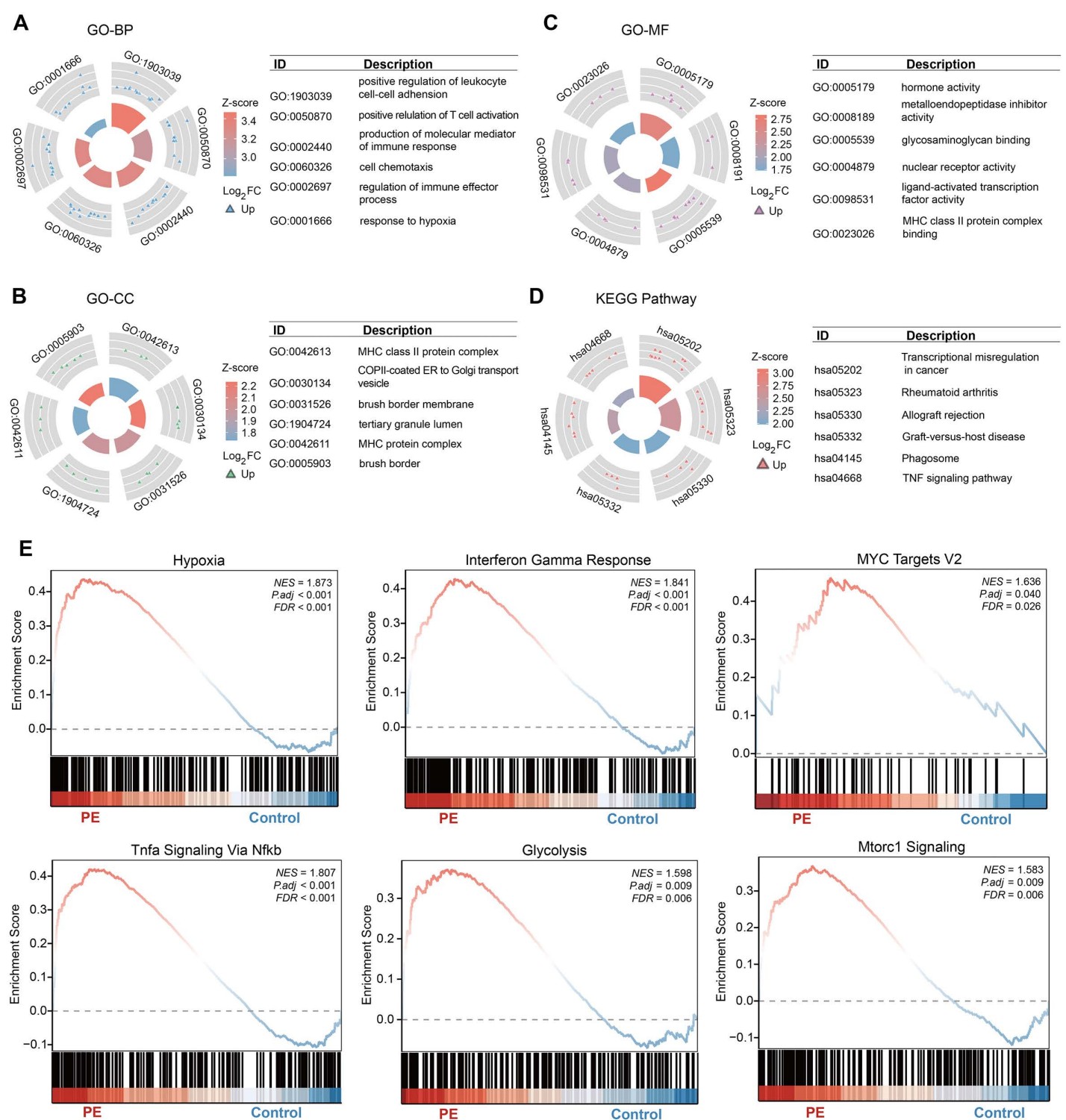

**Fig 3. Functional enrichment analysis. (A-C)** The circle plots show the upregulated DEGs enriched in the BP, CC, and MF GO categories, respectively. **(D)** KEGG enrichment analysis of DEGs. **(E)** Gene set enrichment analysis (GSEA) plots showing pathways enriched in the PE (left) and control (right) groups. NES, normalized enrichment score; FDR, false discovery rate.

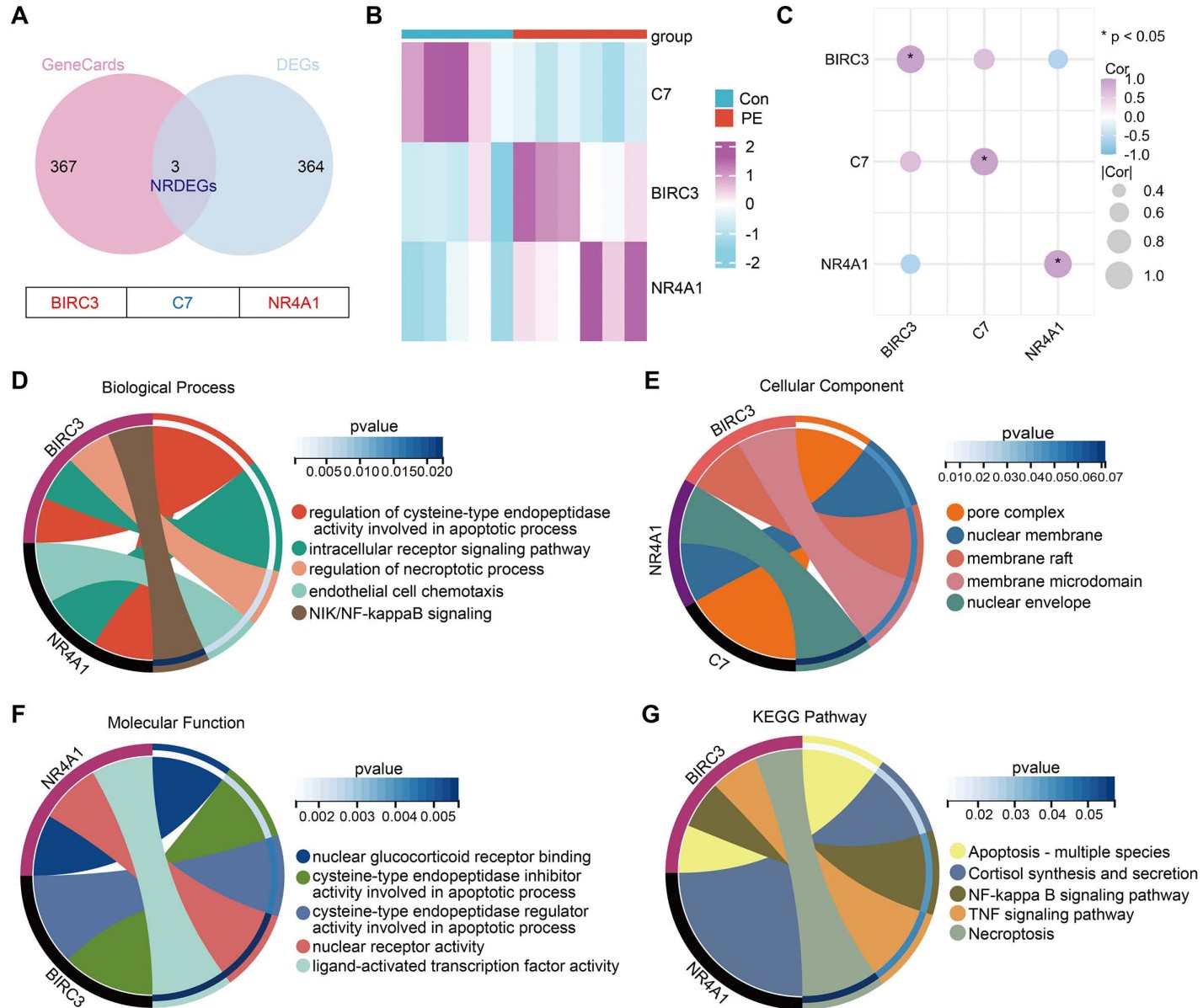

**Fig 4. Screening of necroptosis-related DEGs (NRDEGs) and enrichment analyses. (A)** Venn diagram of overlapping genes between the GSE66273 dataset and necroptosis-related gene sets. **(B)** Heatmap showing the expression levels of the 6 NRDEGs in each sample. **(C)** Spearman correlation among 6 NRDEGs. **(D-G)** Gene functional enrichment analysis of 6 NRDEGs. The cluster tree represents GO-BP, GO-CC, GO-MF, and KEGG, respectively. **(H)** Enrichment heatmap showing the enriched genes associated with BPs, CCs, MFs, and KEGG pathways.

**Table 1. Information of three differentially expressed genes related to necroptosis.**

| Gene name | Description | Log Fold Change | Significance |
|---|---|---|---|
| BIRC3 | Baculoviral IAP Repeat Containing 3 | 1.059925521 | Up |
| NR4A1 | Nuclear Receptor Subfamily 4 Group A Member 1 | 1.262595761 | Up |
| C7 | Complement C7 | −1.363117119 | Down |

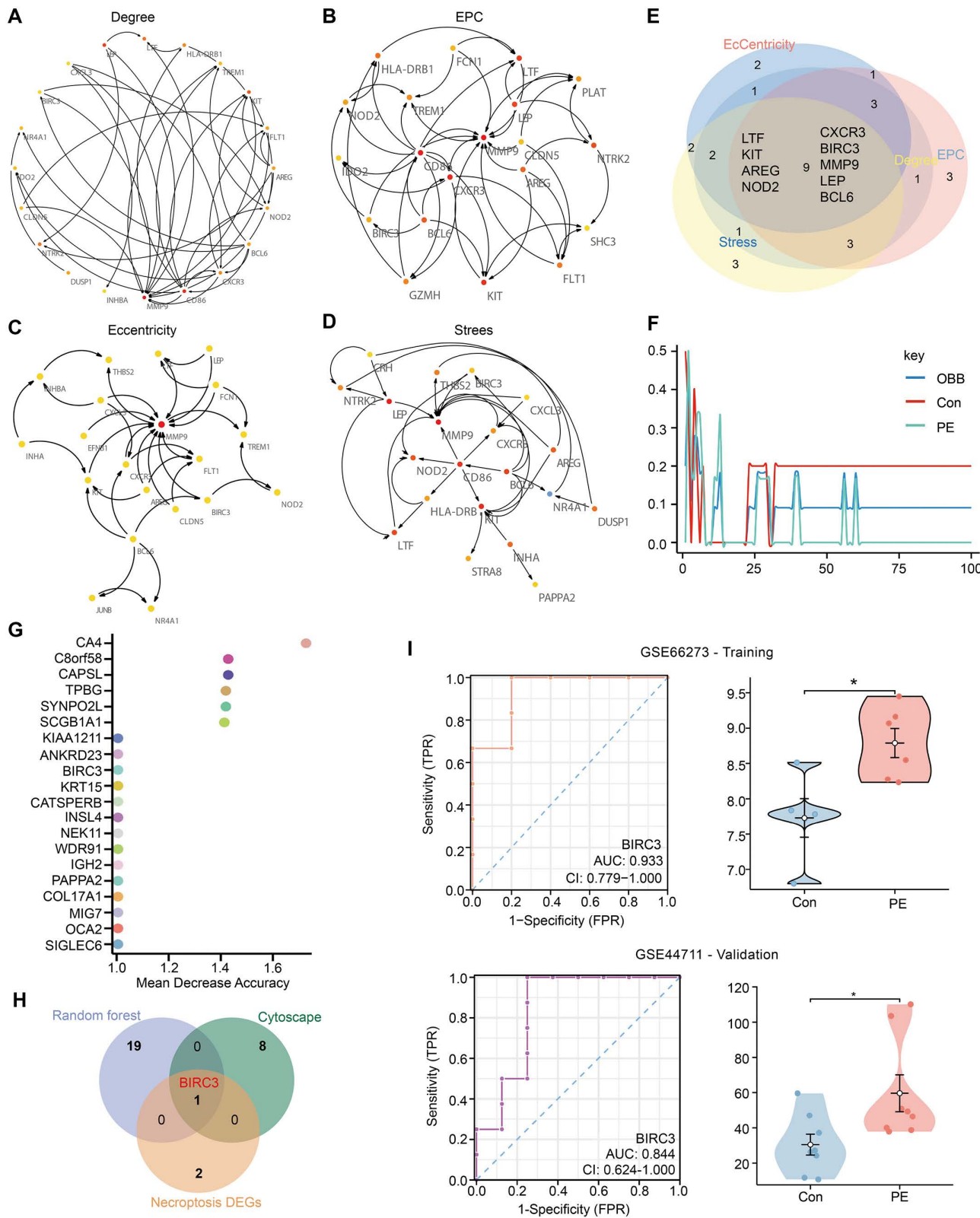

**Fig 5. Identification of hub NRDEGs and their diagnostic value. (A-D)** Hub genes were obtained via CytoHubba on the basis of 200 upregulated DEGs. The figure shows the top 20 hub genes constructed via the degree, EPC, stress, and eccentricity methods. The higher the value is, the redder the

color. **(E)** The intersections of these four algorithms to obtain 9 hub genes. **(F-G)** The RF shows the error and the importance of 20 common genes in PE (GSE66237). **(H)** The Venn map of PE-core gene. **(I)** The expression level and ROC curves of BIRC3.

To evaluate the ability of BIRC3 to distinguish between PE samples and control samples, ROC analysis was conducted on BIRC3 in the training set and validation set, and the Area Under Curve (AUC) value was calculated. The results showed that the AUC values of BIRC3 in the training set and the validation set were 0.933 and 0.844 respectively (Fig 5I). Next, we verified the expression of BIRC3 in different datasets. The results showed that the expression trend of BIRC3 was consistent in both the training set and the validation set, with an up-regulation phenomenon (Fig 5I). The successful external validation confirms that the diagnostic signal of BIRC3 for PE is robust and not an overfitting artifact, though these findings remain preliminary yet promising.

### 3.4 In silico immune infiltration analysis

Inflammatory immune cell infiltration is closely related to the occurrence and development of PE [29]. Immune infiltration was estimated via two different algorithms. We first analyzed the degree of immune infiltration in different groups via ssGSEA and found that the phenotypic characteristics of 24 infiltrating immune cells differed between the control and PE groups (Fig 6A and 6B). There were significant differences in 3 types of immune cell infiltration between the two groups ($P < 0.05$), the level of macrophage, Tcm, and TFH were significantly higher in the PE compared to control group (Fig 6A). Further analysis revealed multiple correlations between immune cells (Fig 6C). Then, immune, stromal, and microenvironmental cell gene signatures were estimated via xCell. It uses the specific markers of each type of immune cell as a gene set to calculate the enrichment score of each type of immune cell in each sample and infer the infiltration of immune cells in a single sample. As shown in Fig 6D, the infiltration levels of various immune cell types, such as CD4 + T cells, CD8 + T cells, macrophages, etc., have undergone significant changes, and these changes may be related to the immune regulatory mechanism of PE. In addition, all the samples were processed with xCell to compute the immune scores and stroma scores. The results revealed a redistribution of immune cells in PE, and the changes in the proportions of these cell types may significantly impact the pathological processes of the condition (Fig 6E and 6F). iDC and T helper cells showed a positive correlation with BIRC3, while macrophages showed negative correlation with BIRC3 (Fig 6G).

To assess the robustness of the immune infiltration patterns, we performed a sensitivity analysis using CIBERSORTx (S1 Fig). We noted some inconsistencies in the estimated abundances of specific immune cell subsets between CIBERSORTx, ssGSEA, and xCell. This variability underscores the challenge and inherent uncertainty of applying computational deconvolution tools to placental transcriptomic data. Nonetheless, a consistent trend suggesting immune dysregulation in PE was observed across methods.

Overall, the core issue in PE is a generalized imbalance and dysfunction of the placental immune microenvironment, rather than a change in one specific, precisely quantifiable immune cell type.

### 3.5 Single-cell analysis and subcellular localization of BIRC3

Numerous public resources are used to explore gene expression in various tissues. We examined the protein expression of BIRC3 in 44 different tissues on the basis of knowledge-based annotation from the HPA [30]. The analysis revealed that BIRC3 was expressed in the placenta samples (Fig 7A). Moreover, we confirmed BIRC3 expression via the IHC results provided by the Human Protein Atlas database (Fig 7B). To determine the specific locations of BIRC3 in cells, we predicted the subcellular localization of BIRC3 using the COMPARTMENTS database. The results showed that BIRC3 was mainly distributed in the cytosol and nucleus (Fig 7C). To determine which cell types in the placenta expressed this hub NRDEG, UMAP plot and bar chart were used to visualize the RNA expression in the single-cell clusters identified in

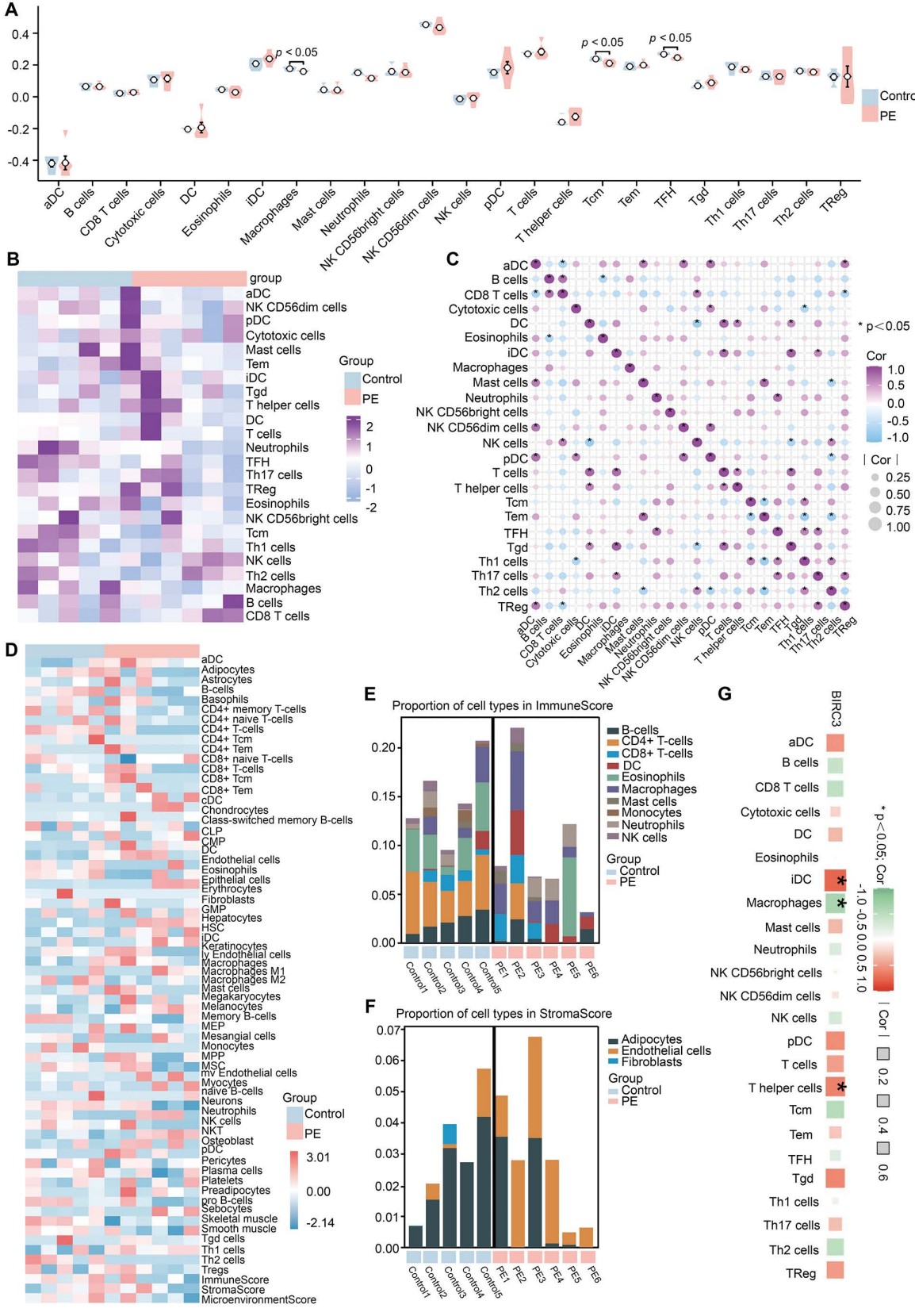

**Fig 6. Infiltration of immune cell types compared between the PE and control groups. (A)** The violin plot shows the proportions of 24 immune cells in the PE group compared with those in the control group, based on the ssGSEA algorithm; red represents the PE group, and blue represents the control group. **(B)** Heatmap of the abundances of 24 immune cell infiltrates based on the ssGSEA algorithm. **(C)** Immune cell correlation graph based on the ssGSEA algorithm. **(D)** Heatmap of the abundances of 64 infiltrating immune cells, based on the xCELL algorithm. **(E-F)** The bar plot shows the proportions of cell types in the ImmuneScore and StromaScore. **(G)** Correlation analysis of BIRC3 and immune cells.

placental tissue. We found that BIRC3 was highly expressed in Hofbauer cells, T cells and immune cells (Fig 7D). Next, we analyzed the data of GSE173193 to further analyze the characteristics between PE and the control group (n = 2). After quality control, a total of 22,976 cells were retained for downstream analysis, including 10,185 in the control group and 12,791 in the PE group. Unsupervised clustering through UMAP found 10 different cell chopsticks (Fig 7E and 7F). Each cluster was annotated based on established marker genes (Fig 7G and 7H and S3 Table): syncytiotrophoblasts (STB; CGA, CYP19A1), neutrophils (Neutro; S100A8, S100A9), cytotrophoblasts (CTB; PAGE4, XAGE3, GSTP1), monocytes/macrophages (Mono/Mac; C1QA, SERPINB2), NK/T cells (NK/T; LYZ, GZMB, NKG7), extravillous trophoblasts (EVT; PAPPA2, AOC1), erythroid cells (Erythro; HBA1, HBB), and fibroblasts (Fib; DLK1, DCN, LUM). The clear separation of these clusters in UMAP space reflects substantial transcriptional heterogeneity among placental cell types. The clear separation of clusters in UMAP space reflects substantial transcriptomic heterogeneity among placental cell types. Comparative analysis revealed significant compositional shifts in PE, including dysregulated proportions of CTB, Mono/Mac, Neutrophils, and EVT (Fig 7I). Critically, cell-type-specific assessment identified BIRC3 expression predominantly in immune lineages, particularly in NK/T cells (Fig 7J and 7K). Moreover, the proportion of BIRC3-positive cells was significantly elevated in PE placentas compared to controls (Fig 7L, Chi-square test, $P < 0.001$), underscoring its potential role in immune dysregulation in PE.

### 3.6 GSEA analysis of BIRC3

To determine the overall trend of signaling pathways related to key gene, single-gene GSEA enrichment analyses were conducted for BIRC3. The pathways associated with the development of PE, such as VEGF signaling pathway, apoptosis, oxidative damage response, HIF1 pathway, tolllike receptor signaling pathway, and complement systems, were identified showing an overall upregulation trend (Fig 8A); while tryptophan catabolism, hormone ligand binding receptors, regulation of FZD by ubiquitination, DNA replication, citric acid cycle TCA cycle and cell cycle checkpoints showed an overall downregulation trend (Fig 8B).

### 3.7 The molecular regulatory networks of BIRC3

TF–gene interaction data were collected via NetworkAnalyst. We included only those interactions that met all of the following stringent criteria: a peak intensity signal < 500, a regulatory potential score < 1, and minimal network connectivity. The network consisted of 6 nodes and 5 edges (Fig 8C). Among them, 5 TFs independently regulate BIRC3, including DEK, REL, NFKB1, HDAC1 and RELA. The miRNA–gene network was also obtained via NetworkAnalyst, which comprised 4 nodes and 3 edges (Fig 8D). BIRC3 is regulated by 3 miRNAs, including hsa-miR-375, hsa-miR-34a-5p, and hsa-miR-98-5p. These silico analysis revealed that a unique and intricate regulatory network may impact hub gene expression.

Studying the interaction between RBPs and RNA is key to exploring the function of RNA. We used the ENCORI database to predict RBP-BIRC3 interactions and visualize the results through a Sankey plot (Fig 8E). The RBP–mRNA interaction network consists of 1 mRNAs (BIRC3), 22 RBP molecules, a total of 22 pairs of mRNA–RBP interaction relationships, and specific mRNA–RBP interaction relationships. Remarkably, the ELAVL1 gene was thought to facilitate cell apoptosis in placental trophoblast cells [31].

These computationally predicted networks are presented as a resource for generating testable hypotheses regarding the upstream regulation of BIRC3.

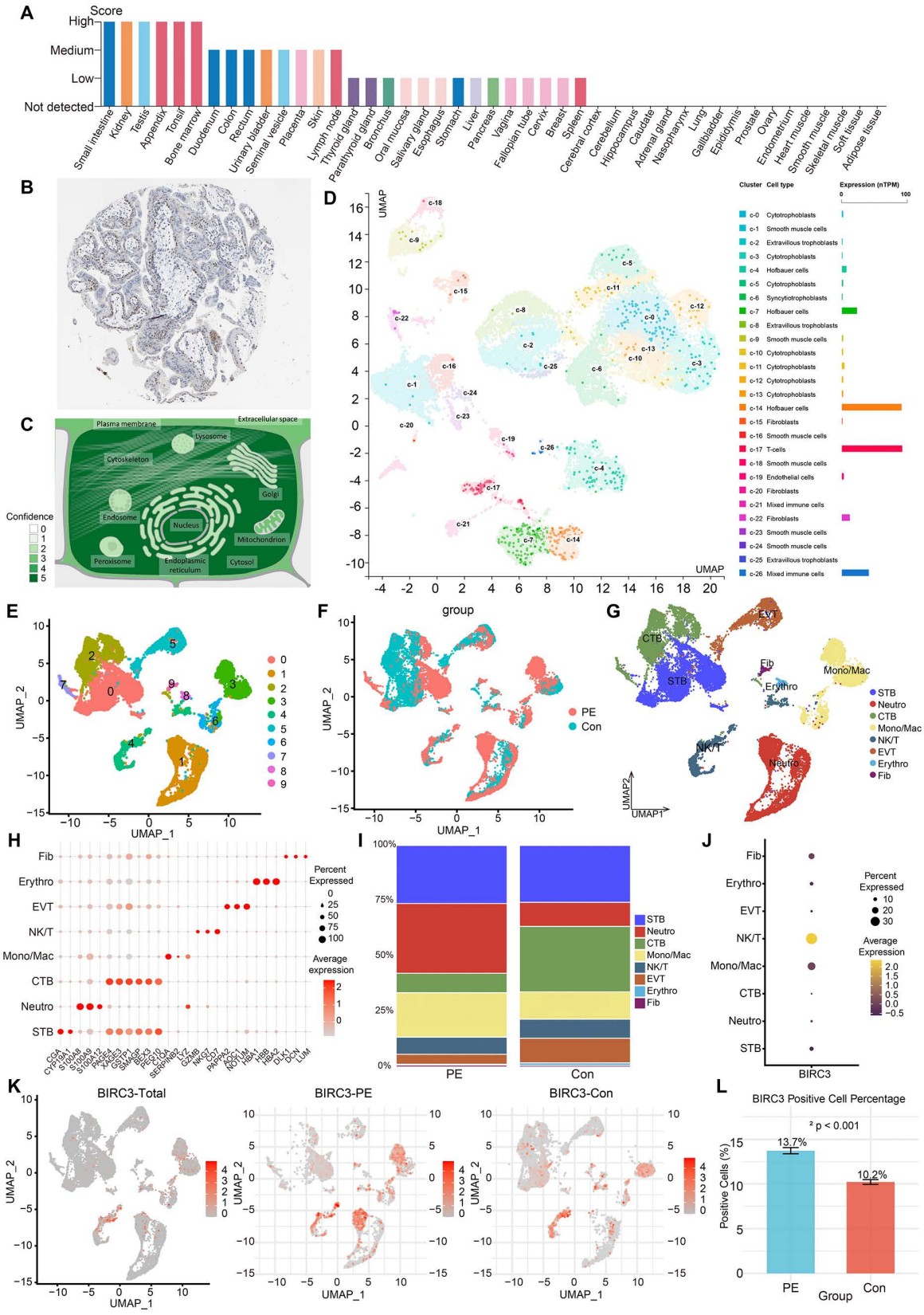

**Fig 7. Single-cell expression analysis and subcellular localization of BIRC3 in tissues and cells. (A)** BIRC3 protein expression overview. **(B)** The immunohistochemical data of BIRC3 in the placenta were analyzed via the HPA database. **(C)** BIRC3 subcellular localization. **(D)** The expression of BIRC3 in the single-cell clusters identified in the placenta visualized by a UMAP plot (HPA database). Each dot corresponds to a cell, and each color corresponds to a cell type. **(E-G)** UMAP representation of the different 10 cell clusters (GSE173193). **(H)** The distribution of cell proportions in different groups. **(I)** Expression map of BIRC3 in different groups.

## 3.8 Identification the candidate drugs and molecular docking

As an exploratory analysis to bridge our biomarker discovery with potential therapeutic strategies, we performed in silico drug prediction and molecular docking. We screened several common drugs (Aspirin, Melatonin, Metformin, Curcumin, Baicalin, Cianidanol, Mangostin, Withanolide D) using molecular docking. The results showed that all compounds effectively bound to BIRC3. Among them, Withanolide D exhibited the strongest binding affinity with the lowest binding energy of −11.0 kcal/mol, followed by Baicalin (−9.0 kcal/mol) and Mangostin (−8.3 kcal/mol) (Fig 8F). It is crucial to emphasize that these findings are derived purely from computational predictions and provide only hypothetical insights. They serve to prioritize these compounds for future experimental validation but do not confirm pharmacological activity.

## 4. Discussion

PE is a severe pregnancy disorder that poses significant risks to both maternal and neonatal health [32]. The deleterious consequences of PE cannot be understood, as it is a leading cause of maternal mortality and morbidity worldwide. However, the underlying mechanisms of PE remain complex and multifaceted. Early diagnosis is particularly difficult because of the lack of effective markers. This lack of understanding highlights the urgent need for a more thorough exploration of the pathogenesis of PE and the discovery of dependable biomarkers for its early detection and accurate diagnosis. This study presents the integrated multi-omics analysis implicating necroptosis in the pathogenesis of PE. By integrating bulk transcriptomics, single-cell RNA sequencing, immune microenvironment profiling, and computational drug screening, we identified BIRC3 as a critical necroptosis-related biomarker and therapeutic target for PE. Our findings bridge a significant knowledge gap by linking regulated necrotic cell death to placental dysfunction and immune dysregulation in PE, moving beyond the traditional focus on apoptosis.

Immune cell dynamics, particularly the presence and activation of specific cell types, such as macrophages, natural killer cells, B cells and T cells, are crucial for maintaining a balanced immune environment during pregnancy [33]. In our study, we performed immune infiltration analysis of placental tissues from the PE and non-PE groups. Our findings revealed that PE patients exhibited a greater degree of infiltration by numerous immune cell types, accompanied by notable increases in the levels of macrophages, T helper cells, and TFH cells within the PE group. Using the xCELL algorithm, we calculated stromal scores and immune scores of PE samples. The results revealed that macrophages and endothelial cells are the primary immune cells according to the immune score and stromal score, respectively. This finding is consistent with the hypothesis that an M1/M2 macrophage imbalance may intensify inflammatory processes and endothelial dysfunction [34,35], which could ultimately result in unfavorable pregnancy outcomes. Crucially, single-cell resolution revealed BIRC3's specific enrichment in placental immune lineages—particularly Hofbauer cells, T cells, and B cells, further supports BIRC3's involvement in immune-mediated placental injury. This localization aligns with its known role in modulating inflammatory signaling and cell survival pathways (e.g., NF-κB, TNF) [36,37]. These findings suggest that BIRC3 may act as a molecular nexus integrating necroptotic signaling with aberrant immune activation in PE pathogenesis.

The consistent upregulation and robust diagnostic performance of BIRC3 across two independent cohorts (AUC = 0.933 in GSE66273 and 0.844 in GSE44711) strongly support its potential as a biomarker for PE. While the AUC in the validation set was slightly lower—as typically expected when moving from discovery to validation—it remains highly promising. Functionally, our study employs an in silico approach to propose a putative regulatory model for BIRC3. Interactions with key transcription factors (NFKB1, RELA, REL), miRNAs (miR-34a-5p, miR-375, miR-98-5p), and RNA-binding proteins (including ELAVL1, known to influence trophoblast apoptosis) indicate complex post-transcriptional control. Their primary

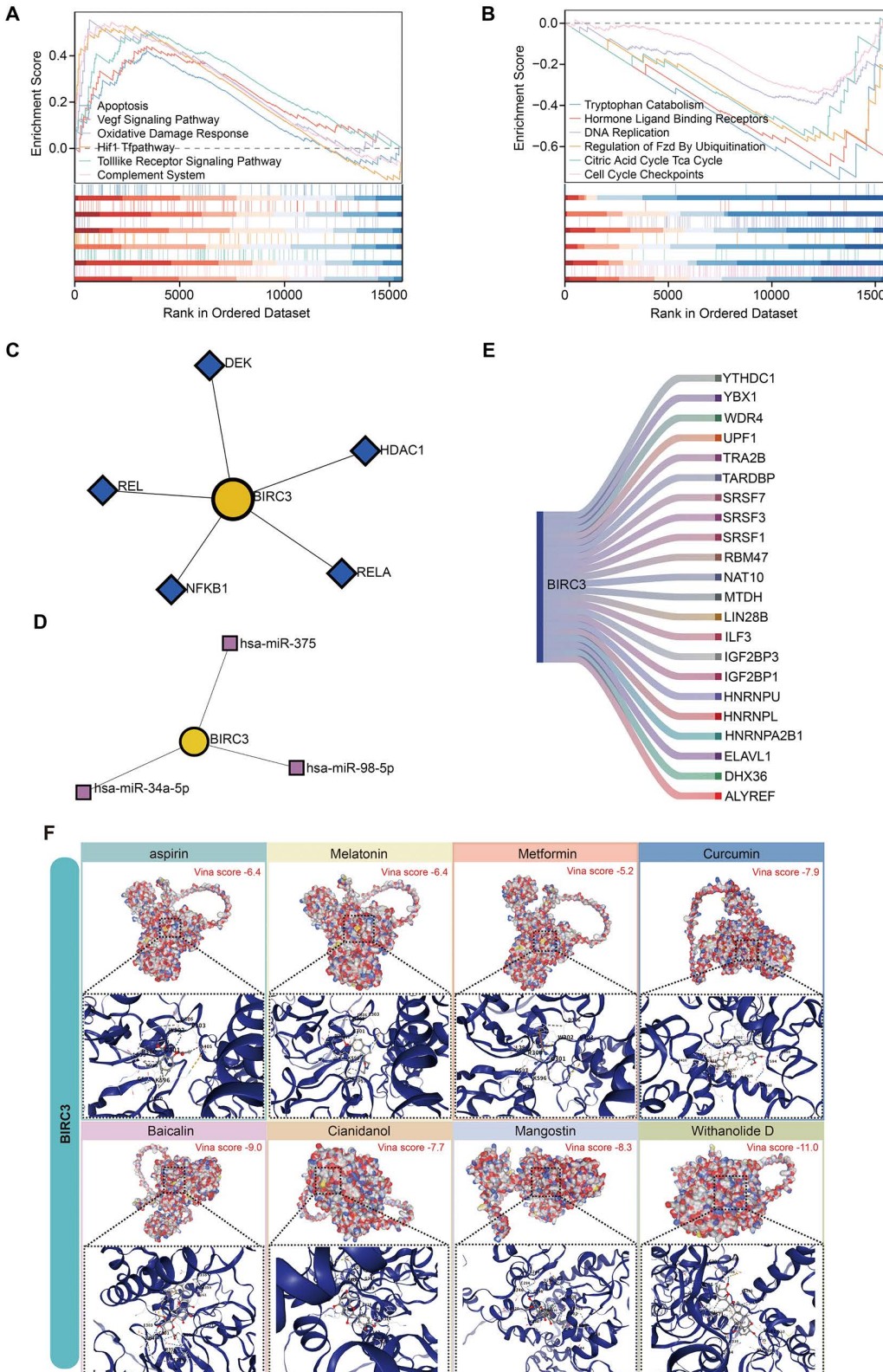

**Fig 8. Molecular regulatory network of BIRC3 and molecular docking. (A-B)** Single-gene GSEA enrichment plots of BIRC3 (typical pathways for up and down regulation). **(C)** TF-Gene regulatory network. The yellow circle represent BIRC3, and the blue rhombuses represent TFs. **(D)** miRNA-gene

regulatory network. The yellow circle represent BIRC3, and the pink squares represent miRNAs. **(E)** RBP-BIRC3 regulatory network. **(F)** Visualization of small molecule and protein binding pattern molecules.

value lies in hypothesis generation, providing a focused set of candidate regulators. GSEA further highlighted BIRC3's association with pathways critical to PE progression: VEGF signaling, hypoxia (HIF1), inflammation, platelet activation, and coagulation cascades were upregulated, while metabolic pathways (tryptophan catabolism, TCA cycle) were suppressed. This signature underscores BIRC3's potential role in driving placental inflammation, vascular dysfunction, and impaired remodeling – hallmarks of PE.

As an exploratory extension of our findings, we employed structure-based virtual screening to identify natural compounds as potential candidates for targeting BIRC3. Three compounds- Withanolide D (predicted binding energy: −11.0 kcal/mol), Baicalin (−9.0 kcal/mol), and Mangostin (−8.3 kcal/mol)-demonstrated favorable theoretical binding affinities in our molecular docking analysis. Notably, these compounds have documented biological activities in existing literature that are relevant to PE pathophysiology: Withanolide D has shown endothelial-protective potential [38]. Baicalin possesses established anti-inflammatory and immunomodulatory properties [39]. Mangostin (−8.3 kcal/mol) can attenuate Ang II-induced hypertension and reverse vascular remodeling, potentially by balancing the ACE/Ang II/AT1R axis and the ACE2/Ang-(1–7)/MasR axis [40]. While the predicted binding energies for all three compounds exceed the commonly referenced threshold for theoretical biological activity (<−7.0 kcal/mol), it is important to emphasize that these results remain preliminary and computationally derived. The docking analysis serves primarily as a hypothesis-generating tool, suggesting that these compounds warrant further investigation as potential BIRC3 modulators. Their documented bioactivities in vascular and inflammatory processes align with key aspects of PE pathogenesis, suggesting potential multi-pathway synergy; however, this conceptual alignment requires experimental validation (e.g., in vitro binding assays, functional studies in relevant cell models).

This study has limitations. First, the initial DEG screening employed a nominal p-value without multiple testing correction, which, while reducing false negatives in our small discovery cohort, increases the potential for false positives. Future studies should prioritize validation in large, independent cohorts with sufficient statistical power to apply stringent FDR correction while retaining sensitivity. Secondly, immune infiltration was assessed using computational deconvolution approaches not optimized for placental tissue, and results varied across algorithms, necessitating cautious interpretation of cell proportion estimates. Thirdly, functional studies using *in vitro* or *in vivo* models are further needed to definitively confirm the mechanistic contribution of BIRC3 to placental dysfunction and necroptosis in PE pathogenesis. Furthermore, the current analysis also lacks spatial context; future investigations employing spatial transcriptomics could elucidate BIRC3 expression dynamics within specific placental microenvironments, particularly at the critical maternal-fetal interface. Finally, while molecular docking identified promising natural compounds (Withanolide D, Baicalin, Mangostin) as potential therapeutics, their biological efficacy, specificity for BIRC3 in placental tissue, and safety profiles during pregnancy remain untested and necessitate rigorous experimental validation in PE-relevant systems.

## 5. Conclusion

This comprehensive investigation identifies necroptosis as a newly recognized pathogenic mechanism in PE and suggests BIRC3 as both a diagnostic biomarker and a therapeutic target. Furthermore, the anticipated natural compounds present promising opportunities for the development of targeted therapies for PE. Subsequent research is warranted to confirm these findings and to further examine the role of BIRC3 in clinical risk assessment and the advancement of drug therapies.

## Supporting information

**S1 Fig. Analysis of immune cell infiltration using CIBERSORTx.**
(TIF)

**S1 Table. Necroptosis related gene set derived from GeneCards database.**
(XLSX)

**S2 Table. Differentially expressed genes in SCI.**
(CSV)

**S3 Table. Information on marker genes of each subpopulation in single-cell sequencing data.**
(CSV)

## Acknowledgments

We thank the Xiantao platform (www.xiantao.love) and Home for researchers platform (www.home-for-researchers.com) for bioinformatics analysis and data generation. We also would like to thank the GEO, Cytoscape, GeneCards databases for the availability of the data.

## Author contributions

**Conceptualization:** Guili Shi.

**Data curation:** Qingxia Lin, Youhong Kang.

**Formal analysis:** Qingxia Lin, Youhong Kang.

**Methodology:** Peifeng Huang.

**Validation:** Qingxia Lin.

**Visualization:** Peifeng Huang.

**Writing – original draft:** Qingxia Lin, Peifeng Huang, Guili Shi.

**Writing – review & editing:** Qingxia Lin, Yanfeng Lu, Guili Shi.

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
