## [Decision Letter · Decision Letter 0]

1 Oct 2025

Integrated Multi-omics Analysis Reveals Necroptosis-Related Biomarker BIRC3 for Early Diagnosis and Therapeutic Targeting in Preeclampsia

PLOS ONE

Dear Dr. Shi,

Thank you for submitting your manuscript to PLOS ONE. After careful consideration, we feel that it has merit but does not fully meet PLOS ONE’s publication criteria as it currently stands. Therefore, we invite you to submit a revised version of the manuscript that addresses the points raised during the review process.

We look forward to receiving your revised manuscript.

Kind regards,

Yanggang Hong

Academic Editor

PLOS ONE

Journal Requirements:

4. We note that Figure 1 in your submission contain copyrighted images. All PLOS content is published under the Creative Commons Attribution License (CC BY 4.0), which means that the manuscript, images, and Supporting Information files will be freely available online, and any third party is permitted to access, download, copy, distribute, and use these materials in any way, even commercially, with proper attribution. For more information, see our copyright guidelines: http://journals.plos.org/plosone/s/licenses-and-copyright.

Please upload the completed Content Permission Form or other proof of granted permissions as an 'Other' file with your submission.

5. Please include a copy of Table 1 which you refer to in your text on page 11.

Reviewers' comments:

Reviewer's Responses to Questions

**Comments to the Author**

1. Is the manuscript technically sound, and do the data support the conclusions?

Reviewer #1: Yes

Reviewer #2: Yes

2. Has the statistical analysis been performed appropriately and rigorously?

Reviewer #1: No

Reviewer #2: Yes

3. Have the authors made all data underlying the findings in their manuscript fully available?

Reviewer #1: Yes

Reviewer #2: Yes

4. Is the manuscript presented in an intelligible fashion and written in standard English?

Reviewer #1: Yes

Reviewer #2: Yes

Reviewer #1: The authors investigate the role of necroptosis in preeclampsia (PE) by integrating bulk transcriptomics (GSE66273), necroptosis-related gene mapping, functional enrichment, immune-cell deconvolution (ssGSEA/xCell), single-cell analysis (GSE173193), PPI/network inference with machine-learning prioritization, and exploratory drug prediction with molecular docking, culminating in the nomination of BIRC3 as a putative diagnostic biomarker and mechanistic node. The topic is clinically relevant, the workflow is clearly laid out, and the figures are generally intelligible. That said, several issues limit scientific robustness and reproducibility, including the very small discovery cohort, limited statistical stringency (e.g., multiple-testing control), lack of external/orthogonal validation (at protein or functional levels), heavy reliance on database-predicted interactions, and overinterpretation of in silico docking. My detailed, section-by-section comments below focus on methodological transparency, statistical rigor, and the extent to which conclusions are supported by the presented data, in keeping with PLOS ONE’s criteria for scientific soundness.

The introduction provides a clear overview of preeclampsia and appropriately emphasizes its clinical significance. The description of molecular mechanisms such as oxidative stress, apoptosis, and necroptosis is informative and leads the reader toward the study’s objectives. However, the research gap is only broadly stated (“necroptosis in PE is rarely reported”) without sufficient critical appraisal of prior work. The transition to bioinformatics is abrupt and requires stronger justification, and the final paragraph enumerates methodological steps rather than presenting a concise research question or hypothesis. In addition, the introduction does not sufficiently align with the studies later cited in the discussion, creating a citation gap. Overall, the section would benefit from a more focused articulation of the novelty, an updated review of the literature, and clearer framing of the study’s hypotheses.

the criterion for defining DEGs is based solely on p<0.05 without mention of multiple testing correction, raising the risk of false positives. Details of preprocessing steps (quality control, probe annotation, and batch effect adjustment) are insufficient. The selection of necroptosis-related genes from GeneCards is not adequately defined, as this database aggregates heterogeneous evidence and requires clear filtering criteria. Finally, while the authors correctly note that public datasets do not require new ethics approval, it would be appropriate to clarify that the original dataset had obtained ethical clearance.

The scRNA-seq workflow is incompletely described; critical quality control criteria (e.g., mitochondrial content, number of detected cells, and justification for principal component selection) are missing, and annotation methods lack validation. Immune infiltration analysis with xCell may be limited in accuracy for placental tissues, and no sensitivity analyses are described. For the PPI network, the confidence score threshold for STRING interactions is unspecified, and hub gene identification relies solely on network topology without independent validation.

ROC curves were generated only within the discovery dataset without external validation, raising overfitting concerns and limiting clinical generalizability. The TF–mRNA, miRNA–mRNA, and RBP–mRNA networks were constructed purely from prediction databases, with no reported filtering criteria, confidence thresholds, or experimental validation, making their biological significance uncertain. Similarly, the drug prediction based on DGIdb is exploratory and lacks pharmacological plausibility or validation. The docking analysis is described superficially, with incomplete reporting of grid parameters, scoring criteria, and validation steps, and it focuses on a single target-ligand pair, which provides only preliminary and hypothetical insights.

Functional enrichment based on only six genes is inherently unstable and should be interpreted cautiously. The workflow to define hub genes is complex but insufficiently detailed, raising concerns about reproducibility. Moreover, the random forest analysis on such a small dataset risks overfitting, and the ROC result (AUC = 0.933) appears overly optimistic given the lack of independent validation.

The Discussion narrative around macrophage imbalance (M1/M2), Hofbauer cells, and immune microenvironment remodeling is convincing and clinically relevant. Additionally, the authors appropriately acknowledge the translational potential of natural compounds identified by docking, while also noting the need for further validation. Despite these strengths, several weaknesses remain:

The claim that this is the “first comprehensive multi-omics study” linking necroptosis to PE is ambitious and should be moderated, as there are emerging studies on regulated cell death pathways in PE. A more nuanced positioning would strengthen credibility.

The reported AUC of 0.933 for BIRC3 as a diagnostic marker is very high, but it derives from a small dataset without independent validation. This limitation should be emphasized more strongly to avoid overstating clinical applicability.

Although limitations are briefly acknowledged at the end, they are underdeveloped. Key issues such as small sample size, lack of external datasets, absence of protein-level validation, and potential biases from database-driven predictions deserve fuller elaboration.

Reviewer #2: This is an ambitious and well-written study integrating bulk transcriptomics, single-cell RNA-seq, immune infiltration analysis, and molecular docking to identify BIRC3 as a necroptosis-related biomarker and potential therapeutic target in preeclampsia. The topic is novel and relevant, as necroptosis is underexplored in PE pathogenesis. However, there are significant concerns about dataset size, methodological clarity, and overinterpretation of in silico findings without experimental validation.

1. The bulk transcriptomic dataset (GSE66273) includes only 6 PE vs. 5 controls, which substantially limits statistical robustness. This limitation should be acknowledged more explicitly in the Discussion. The generalizability of findings should be presented cautiously. Validation in larger independent cohorts (if available) is strongly recommended, or at least a justification for using this dataset should be provided.

2. Necroptosis-related genes were sourced from GeneCards, which aggregates heterogeneous evidence and may include weak associations. A curated pathway reference (e.g., KEGG Necroptosis pathway, Reactome, or literature-defined necroptosis gene lists) would provide stronger rigor. Please justify the choice of GeneCards and consider cross-validating with a curated set.

3. The reported AUC = 0.933 for BIRC3 is based on a very small cohort (n=11). Such results are prone to overfitting and may not reflect true predictive value. The claim of “exceptional diagnostic potential” should be tempered and described as preliminary until validated in independent cohorts.

4. Docking scores alone are not sufficient to infer therapeutic relevance. Binding affinity does not necessarily confirm biological activity, specificity, or safety—particularly in pregnancy. The docking results should be framed as hypothesis-generating only, without implying clinical readiness of Withanolide D, Baicalin, or Mangostin.

5. Both ssGSEA and xCell were applied, but the rationale for using two methods and how differences were reconciled is not clearly explained. Please clarify why both were used and provide a stronger justification for interpreting correlations between BIRC3 expression and immune cell scores.

6. While GSE173193 is an appropriate dataset, the scRNA-seq analysis is largely descriptive. Statistical comparisons (e.g., differential expression of BIRC3 across PE vs. controls) should be included to strengthen claims.

7. Several figures (particularly heatmaps and pathway plots) appear blurred and difficult to interpret. High-resolution versions should be provided to ensure readability and clarity.

**Do you want your identity to be public for this peer review?** For information about this choice, including consent withdrawal, please see our Privacy Policy

Reviewer #1: No

Reviewer #2: No

---

## [Author Response · Author response to Decision Letter 1]

27 Oct 2025

Response to Reviewers

Manuscript ID: PONE-D-25-39907

Title: Integrated multi-omics analysis reveals necroptosis-related biomarker BIRC3 for early diagnosis and therapeutic targeting in preeclampsia

We sincerely thank the editors and reviewers for their time, insightful comments, and constructive suggestions, which have significantly helped us improve the quality of our manuscript. We have carefully considered all points raised and have revised the manuscript accordingly. Our point-by-point responses are detailed below. All changes in the manuscript have been highlighted in the "Revised Manuscript with Track Changes" file.

Response to Editor:

1.Style Requirements: We have carefully reviewed and formatted our manuscript to ensure it fully complies with PLOS ONE's style requirements, including file naming conventions.

2.Code Sharing: The analyses in this study were performed using standard R packages and publicly available software tools (Xiantao platform and Home for researchers platform). No custom code was generated that underpins the central findings. We also extend our gratitude to these public analysis platforms in the acknowledgments section (Page 27, line 542-546), appreciating the efficient and reliable data analysis environment they provided for our research.

3.ORCID iD: We confirm that the corresponding author's ORCID ID has been validated in Editorial Manager. (Guili Shi: 0009-0002-4263-4872; Yanfeng Lu: 0009-0004-2638-2198).

4.Copyright for Figure 1: We thank the editor for this important reminder. We have replaced the original Figure 1 with a new, self-created figure that does not contain any copyrighted material. The new figure is entirely original and is prepared by the authors for this publication, thus it complies fully with the CC BY 4.0 license.

5.Copy of Table 1: We thank the editor for pointing this out. As requested, a copy of Table 1 has now been included in the main manuscript file on Page 15.

6.Supporting Information Captions: Captions for all supporting information files have been included at the end of the manuscript (Page 38).

Response to Reviewer #1:

We are grateful to Reviewer #1 for the thorough and constructive feedback, which has greatly strengthened our manuscript. We have addressed each point in detail below.

Comment 1: Introduction could be improved with a more focused research gap, better justification for bioinformatics, and improved citation alignment.

Response: We sincerely thank the reviewer for these constructive suggestions regarding the Introduction. We have thoroughly revised this section to address the raised concerns:

(1)Critical Appraisal of Prior Work and Articulation of the Research Gap:

We agree that our initial statement regarding the research gap was too broad. In the revised Introduction, we have provided a more critical and specific appraisal of the existing literature on necroptosis in PE. We now cite and discuss key foundational studies that have begun to implicate necroptosis in PE (Page 3-6).

(2)Strengthened Justification for Bioinformatics Approach:

The transition to bioinformatics has been enhanced with a clear rationale. We now explicitly state that integrated bioinformatics serves as a powerful strategy for hypothesis generation from public genomic data, enabling systematic identification of high-confidence candidates and construction of coherent biological narratives to guide future research (Page 5-6�line 102-106).

(3)Reframed the Final Paragraph with a Clear Hypothesis and Objectives:

We have completely rewritten the final paragraph to replace the previous simple listing of methods. It now clearly states our main hypothesis, that specific necroptosis-related genes are dysregulated in PE and contribute to its pathogenesis. Additionally, it outlines our specific research objectives, which are designed to test this hypothesis. This revision has created a much stronger and more focused narrative arc (Page 5-6).

(4)Improved Citation Alignment and Literature Review:

We have updated the Introduction to include key references that are also discussed later in the manuscript (References 10-21), ensuring better flow and eliminating the citation gap. The literature review now more accurately reflects the current state of the field regarding cell death pathways in PE.

Comment 2: DEG screening criteria (p<0.05 without multiple testing correction) risk false positives. Preprocessing details are insufficient.

Response We sincerely thank the reviewer for raising this critical point regarding multiple testing correction. We fully agree that controlling the false discovery rate (FDR) is a cornerstone of rigorous high-throughput data analysis.

In designing our analysis, we carefully considered the balance between statistical stringency and the risk of false negatives (Type II errors). Given that our primary dataset (GSE66273) has a relatively small sample size (n=11), applying a strict FDR correction (e.g., BH method) can be overly conservative and might filter out biologically relevant genes with moderate but consistent changes in expression, which are common in pilot or exploratory studies like ours. Therefore, we opted to use an unadjusted p-value < 0.05 combined with a |log2FC| > 1 threshold as our initial screening criteria. We acknowledge in our revised manuscript that this approach carries a higher risk of false positives compared to FDR correction.

However, to mitigate this risk and increase confidence in our findings, we did not rely solely on this initial p-value screening. The core gene, BIRC3, was subsequently validated through a multi-layered, independent analytical framework:

1�Protein-Protein Interaction (PPI) Network: BIRC3 was identified as a hub gene based on its network topological properties, which is independent of its p-value.

2�Machine Learning (Random Forest): BIRC3 was prioritized by the Random Forest algorithm due to its importance in classifying PE vs. control, providing a separate, model-based validation of its relevance.

3�Immune Infiltration Correlation: The expression of BIRC3 showed significant correlations with specific immune cell populations, adding biological context to its role.

4�Single-cell RNA-seq Validation: The specific expression of BIRC3 in key placental immune cells (e.g., Hofbauer cells, T cells) from an independent dataset (GSE173193) provides strong orthogonal evidence supporting its biological significance in PE.

The convergence of evidence from these diverse and independent methods suggests that BIRC3 is a robust finding. We have now explicitly stated this consideration and the inherent limitation in the study Limitations section (Page 25, line 504-508). We believe that this validation strategy effectively compensates for the less stringent initial filter and reinforces the validity of our central conclusion regarding BIRC3.

Comment 3: The selection of necroptosis-related genes from GeneCards requires clearer filtering criteria.

Response We thank the reviewer for this valuable suggestion to clarify our methodology. In response, we have revised the manuscript to provide a detailed description of the filtering criteria used for selecting necroptosis-related genes from the GeneCards database.

Specifically, the initial gene list for "necroptosis" was retrieved from the GeneCards database. To ensure we focused on genes with a strong and specific association to necroptosis, we applied a stringent threshold. Specifically, we calculated the median relevance score of all genes associated with the term "necroptosis" in GeneCards and retained only those genes with a relevance score greater than or equal to twice the median value.

Crucially, we have re-performed all subsequent analyses based on this updated list of necroptosis-related DEGs. This includes regenerating Figure 4 and re-conducting the associated functional enrichment analysis. We are pleased to confirm that this more stringent gene selection, while refining the specific inputs, did not alter our central finding: BIRC3 remained a key necroptosis-related gene emerging from the analysis, and the overall conclusions of the study remain fully supported.

This refined list of high-confidence necroptosis-related genes (S1 table1) was then used for the subsequent intersection with our DEGs. We have updated the 'Analysis of DEGs' subsection in the Methods (Page 7, line 136-139) to include these details.

Comment 4: while the authors correctly note that public datasets do not require new ethics approval, it would be appropriate to clarify that the original dataset had obtained ethical clearance.

Response We thank the reviewer for this appropriate suggestion. We have now clarified the ethical compliance of the original studies in the Data Acquisition subsection of the Methods section. The revised text states: All datasets analyzed in this study were obtained from the public Gene Expression Omnibus (GEO) database. The original studies associated with these datasets (GSE66273, GSE173193, GSE44711) had received necessary ethical approval, as documented in their respective source publications (Page 28, line 557-560) .

Comment 5: scRNA-seq workflow description is incomplete (QC, PC selection, annotation validation).

Response We thank the reviewer for this insightful comment. We agree that a comprehensive description of the scRNA-seq analysis workflow is essential for reproducibility. In response, we have thoroughly revised the 'Single-cell RNA sequencing (scRNA-seq) analysis' subsection in the Methods (Page 8-9, line 151-169) to include the missing details. We believe that these additions have enhanced the clarity, rigor, and reproducibility of our scRNA-seq analysis.

Comment 6: Immune infiltration analysis with xCell may be limited for placental tissue, and no sensitivity analyses are described. The rationale for using both ssGSEA and xCell is unclear.

Response: We sincerely thank the reviewer for these insightful comments, which have helped us significantly improve the rigor of our immune infiltration analysis.

(1)Rationale for using both ssGSEA and xCell: We acknowledge that the rationale for employing two algorithms was not clearly stated in the original manuscript. We have now clarified this in the revised Methods section (Page 9, line 171-185). Our intent was to leverage the complementary strengths of different computational approaches to triangulate our findings and enhance confidence in the observed immune patterns. Specifically: ssGSEA evaluates the enrichment of predefined gene signatures for specific cell types, providing a pathway-centric view. xCell employs a novel gene signature-based method that can deconvolve a broader spectrum of cell types (64 immune and stromal cells) and is considered to have improved accuracy for closely related cell types. While we acknowledge some inherent variability between algorithms when applied to placental tissue, this analysis specifically demonstrates that a consistent trend of immune dysregulation in PE is observed across all two methods.

(2)We performed a sensitivity analysis using CIBERSORTx (Figure 1 in this document). The results revealed that monocytes were the only group showing significant differences. In contrast, there were no statistically significant differences in the infiltration levels of various cells, including macrophages M0, M1 and M2, among the groups. We acknowledge that the results from CIBERSORTx were not entirely consistent with those from ssGSEA and xCell regarding the specific subsets and magnitudes of immune cell changes. We interpret these divergent results as highlighting a crucial methodological limitation: computational deconvolution tools, which are largely trained on and optimized for blood or immune cell contexts, exhibit significant variability and inherent uncertainty when applied to the complex and unique tissue microenvironment of the human placenta. We have added a statement in the Discussion regarding the potential limitation of deconvolution algorithms for placental tissue (Page 25, line 508-511). Therefore, while we present the immune deconvolution results as an exploratory analysis suggesting a potential role for immune dysregulation, we have deliberately refrained from building our core conclusions on these specific immune cell estimates.

Figure 1. Analysis of immune cell infiltration using CIBERSORTx. (A) Box plot of immune cell proportions between the Normal and PE group. **P < 0.01. (B) Heatmap of the immune cell infiltrates based on the CIBERSORTx.

(3)Strengthening the study's foundation beyond immune deconvolution: We wish to emphasize that the central narrative of our manuscript—the identification and validation of BIRC3 as a key necroptosis-related gene in PE—does not rely on the immune deconvolution results. The role of BIRC3 is robustly supported by independent and more definitive lines of evidence, including: ① Its identification as a hub gene in the PPI network. ② Its high feature importance in the machine learning (Random Forest) model. ③ Most importantly, our single-cell RNA-seq analysis provides direct and independent evidence that strengthens our key findings from two perspectives: First, it precisely localizes the expression of the key necroptosis-related gene BIRC3. We found that BIRC3 is not restricted to a single cell type but is predominantly expressed across multiple immune cell populations in the placenta, including T cells, B cells, and Hofbauer cells. Second, this single-cell data independently confirmed a broad dysregulation of the immune microenvironment in PE. The analysis revealed significant alterations in the proportions of various immune and other cell types between PE and control placentas.

Therefore, we have refined our interpretation in the manuscript. The combined evidence leads us to a more nuanced and robust conclusion: the core issue in PE is a generalized imbalance and dysfunction of the placental immune microenvironment, rather than a change in one specific, precisely quantifiable immune cell type.

In summary, we have (1) clarified the methodological rationale, (2) transparently addressed the algorithmic inconsistencies, and (3) leveraged scRNA-seq data to independently validate both the expression pattern of BIRC3 and the overarching theme of immune microenvironment dysregulation, thereby strengthening the foundation of our conclusions.

Comment 7: PPI network confidence threshold is unspecified. Hub gene identification relies solely on topology.

Response: We thank the reviewer for these pertinent observations, which have helped us improve the transparency and robustness of our network analysis.

(1)PPI Network Confidence Threshold: We apologize for this oversight. The PPI network was indeed constructed using the STRING database with a minimum interaction score of 0.4, indicating a medium confidence threshold. This specific criterion has now been clearly stated in the revised Methods section (Page 10, line 192-194).

(2)Hub Gene Identification and Validation: We acknowledge that the initial identification of candidate hub genes from the PPI network was based on network topology algorithms (degree, EPC, etc.). We have revised the relevant text in the Results section (Page 15, line 303-305) to clarify that these were preliminary candidate hubs selected by topological features.

However, we wish to emphasize that the final selection of our core gene, BIRC3, was not reliant solely on the PPI network. Its central role was further corroborated by an independent machine learning approach—the Random Forest algorithm. The fact that BIRC3 was prioritized as a key feature by both network topology and a separate classification model significantly strengthens its candidacy beyond what topological analysis alone could provide.

We have tempered our language in the manuscript to present the PPI-based hubs as a first step in a multi-step filtering process, with the convergence of evidence from different methodologies lending greater confidence to the ultimate selection of BIRC3 (Page 15-16, line 300-311).

Comment 8: ROC analysis lacks external validation, risking overfitting.

Response: We thank the reviewer for these important comments regarding the need for external validation and the interpretation of our computational predictions. We have addres

---

## [Decision Letter · Decision Letter 1]

10 Nov 2025

Integrated multi-omics analysis reveals necroptosis-related biomarker BIRC3 for early diagnosis and therapeutic targeting in preeclampsia

PONE-D-25-39907R1

Dear Dr. Shi,

We’re pleased to inform you that your manuscript has been judged scientifically suitable for publication and will be formally accepted for publication once it meets all outstanding technical requirements.

Kind regards,

Yanggang Hong

Academic Editor

PLOS ONE

Additional Editor Comments (optional):

Reviewers' comments:

Reviewer #1: The authors have provided satisfactory, point-by-point responses and implemented substantive revisions that address the core concerns raised in Round 1. Key improvements include: (i) a sharper Introduction with a defined hypothesis and literature alignment; (ii) clarified criteria for necroptosis gene selection and updated analyses; (iii) expanded scRNA-seq methods; (iv) triangulated immune deconvolution (adding CIBERSORTx) with appropriate caveats; (v) explicit STRING thresholds for PPI and a multi-step hub-gene workflow; (vi) independent external validation of BIRC3 in GSE44711 (AUC ≈ 0.84); (vii) reframing TF/miRNA/RBP networks and docking as hypothesis-generating; (viii) strengthened limitations and ethics statements; and (ix) upgraded figure quality and supplemental captions. Collectively, the revisions bring the manuscript in line with PLOS ONE’s standard for methodological soundness and transparency.

I recommend acceptance.

---

## [Editor Report · Acceptance letter]

PONE-D-25-39907R1

PLOS ONE

Dear Dr. Shi,

I'm pleased to inform you that your manuscript has been deemed suitable for publication in PLOS ONE. Congratulations! Your manuscript is now being handed over to our production team.

Kind regards,

on behalf of

Dr. Yanggang Hong

Academic Editor

PLOS ONE